# A 2D Porous Zinc-Organic Framework Platform for Loading of 5-Fluorouracil

**Liang Qin [1], Fenglan Liang [2],*, Yan Li [1], Jiana Wu [1], Shiyuan Guan [1], Meiyin Wu [1], Shiling Xie [1], Manshi Luo [1] and Deyun Ma [1],***

[1] School of Food and Pharmaceutical Engineering, Zhaoqing University, Zhaoqing 526061, China
[2] College of Life Science, Zhaoqing University, Zhaoqing 526061, China
* Correspondence: liangfl82@126.com (F.L.); mady@zqu.edu.cn (D.M.)

**Abstract:** A hydrostable 2D Zn-based MOF, $\{[Zn(5\text{-PIA})(imbm)]\cdot 2H_2O\}_n$ (**1**) (5-$H_2$PIA = 5-propoxy-isophthalic acid, imbm = 1,4-di(1H-imidazol-1-yl)benzene), was synthesized and structurally characterized. Complex **1** shows good water and thermal stability based on the TGA and PXRD analyses and displays a 2D framework with 1D channels of $4.8 \times 13.8$ and $10.0 \times 8.3$ Å$^2$ along the *a* axis. The 5-fluorouracil (5-FU) payload in activated complex **1** (complex **1a**) is 19.3 wt%, and the cumulative release value of 5-FU at 120 h was about 70.04% in PBS (pH 7.4) at 310 K. In vitro MTT assays did not reveal any cytotoxic effect of NIH-3T3 and HEK-293 cells when the concentration of **1** was below 500 µg/mL and 5 µg/mL, respectively. No morphological abnormalities were observed on zebrafish exposed to complex **1**.

**Keywords:** zinc-organic framework; drug delivery; 5-fluorouracil; MTT assays; zebrafish toxicity

## 1. Introduction

Conventional chemotherapy often requires high doses of anti-cancer drugs due to poor biodistribution, which leads to serious side effects [1]. To address this, it is important to develop delivery systems for drugs that are efficient and novel. Metal-organic frameworks (MOFs) represent a promising class of highly ordered crystalline porous materials, and they have received considerable attention [2–7]. According to recent studies, MOF-based drug carriers are capable of delivering targeted drugs, which makes them an attractive class for drug delivery, mainly because of their capability to carry large amounts of drugs, controlled release of drug molecules within the frameworks, low toxicity, and high efficiency [8–10]. Since Ferey and co-workers reported the first study of MOFs for drug loading and delivery, many similar studies have been reported in recent years. To date, a series of drug molecules, including doxorubicin, cisplatin, topotecan, camptothecin, busulfan, and 5-fluorouracil, have been loaded into MOFs for intracellular delivery and cancer treatment [11]. For example, Lin and coworkers reported the loading of cisplatin in MIL-101 (Fe) in 2009 [12]. The Zhou group reported a water-stable mesoporous material (PCN-333(Al)) used for enzyme encapsulation, which prevented enzymes from aggregation and leaching [13]. ZIF-8 was synthesized with camptothecin encapsulated inside the framework [14]. In addition to preventing substrate leaching, MOFs provide a protective environment for the loaded substrates against external hazards.

Several interactions that affect the controlled release performance of drug molecules in MOFs/drug systems include van der Waals' forces, hydrogen bonds, coordination bonds, and π-π stacking interactions [15]. Furthermore, it has been found that drug loading in MOFs is governed by cage accessibility, while loading capacity depends on the hydrophobicity/hydrophilicity of MOFs and drug molecules [16].The Horcajada group has shown that both MOF structure and drug molecules' hydrophobicity/hydrophilicity could influence the drug release rate. Hydrophobicity and hydrophilicity mismatches could cause drugs to release rapidly.

Furthermore, MOFs' function, biocompatibility, and stability in water solutions are crucial for biomedical applications. Many reported MOFs were constructed by toxic metal ions ($Cr^{3+}$, $Cd^{2+}$, etc.) and other harmful constituents, leading to the potential toxicity of MOF-based drug delivery systems [17]. In addition to their function and biocompatibility, MOFs must also be stable in water solutions for biomedical applications [18]. In many studies, zinc-based MOFs have been shown to be unstable, and when submerged in water or exposed to the air, they lose their structures as well as large surface areas rapidly [19]. For example, MOF-5 is unstable in water or moisture [20]. Clearly, this drawback would greatly limit their applications in drug delivery.

As part of an on-going study related to 5-fluorouracil (5-FU) drug delivery based on zinc-based MOFs [21–24], we report the synthesis and structural characterization of one 2D Zn-based MOF involving carboxylate ligands and N-containing auxiliary ligands and explore their drug delivery properties. Over the past few decades, carboxylate ligands and N-containing auxiliary ligands have been used in the construction of a variety of fascinating MOFs [25].

We focused on a versatile 5-propoxy-isophthalic acid ligand with various coordination modes [26] and an N-containing auxiliary ligand of 1,4-di(1H-imidazol-1-yl)benzene. Herein, a new water-stable 2D zinc-based MOF, [Zn(5-PIA)(imbm)]$_n$ (**1**), has been first synthesized under a mixed-solvent ($H_2O$/ethanol) condition. To date, complex **1** represents the first example constructed by 5-propoxy-isophthalate and 1,4-di(1H-imidazol-1-yl)benzene ligands. As previously reported, 5-fluorouracil (5-FU), a typical anticancer drug, was selected as a model drug due to the channel size of complex **1** is suitable for 5-FU encapsulation. The in vitro cytotoxicity of complex **1** was evaluated by MTT assays. The in vivo toxicity of complex **1** was evaluated using the zebrafish embryo model. The aim of this paper is to explore the encapsulation of complex **1** on 5-fluorouracil and the toxicity results of complex **1** on cells (NIH-3T3 and HEK-293) and zebrafish, respectively, which can enrich the study of drug delivery systems based on the MOFs materials.

## 2. Results and Discussion

### 2.1. The Crystal Structure of Complex **1**

Single-crystal X-ray diffraction results reveal that the titled compound crystallizes in the P21/n space group with the asymmetric unit containing one zinc ion, one 5-PIA anion, and one imbm ligand. Zn1 is four-coordinated by carboxylate oxygen atoms from two different 5-PIA anions and two N atoms from two different imbm ligands, adopting a distorted tetrahedral coordinated sphere (Figure 1A) with the Zn-O, Zn-N distances, and O-Zn-O, O-Zn-N, and N-Zn-N bond angles in the range of 1.9359(17)–2.021(2) Å, and 98.37(8)–121.65(8)°, respectively. The results are in good agreement with other four-coordinated Zn(II) complexes [27]. The 5-PIA anion acts in a bridging-$\mu_2$ mode to link two zinc cations, whereas the imbm ligand acts as a trans-bidentate bridging ligand to connect the two Zn(II) ions (Scheme 1). The Zn(II) cations are connected into a zigzag chain through 5-PIA ligands with a Zn···Zn distance of 9.430 Å (Figure 1B). These chains are further linked into a 2D (4,4) network when the zinc ions and the organic ligands are considered as nodes and linkers, respectively (Figure 2). Each 4-membered ring consists of two 5-PIA anions and two imbm ligands, with channels of 4.8 × 13.8 Å$^2$ and 10.0 × 8.3 Å$^2$ along the *a*-axis.

### 2.2. IR, Thermal, and Chemical Stability

The FT-IR spectrum result (recorded by KBr pellets) of complex **1** is shown in Figure 3. The absorption band at 3625 cm$^{-1}$ may be attributed to the $\nu$(O-H) stretching vibrations from the guest water molecules. The absorption bands at 2964 and 2873 cm$^{-1}$ are attributed to the methyl and methylene groups in 5-PIA ligands, respetively. The stretching bands at 1684 cm$^{-1}$ and 1383 cm$^{-1}$ are due to asymmetric and symmetric C-O-C vibrations.

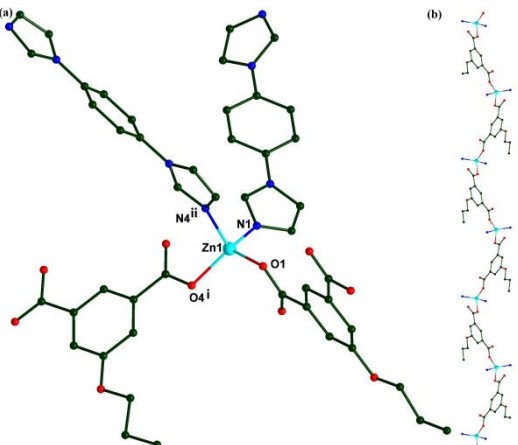

**Figure 1.** (**a**) The coordination environment of Zn(II) ions in complex **1**. (**b**) View of the 1D zigzag chain of complex **1**. Symmetry codes: i = x + 1/2,−y + 1/2,z + 1/2; ii = −x + 5/2,y − 1/2,−z + 3/2. Color codes: Zn, turquiose; O, red; C, dark green; N, blue.

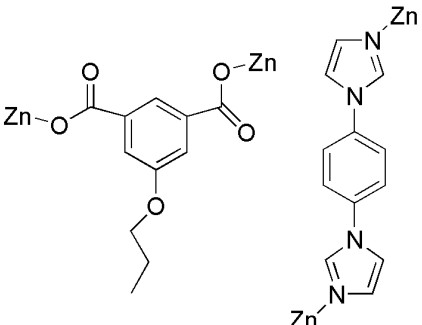

**Scheme 1.** The coordination modes of 5-PIA and imbm ligands in complex **1**.

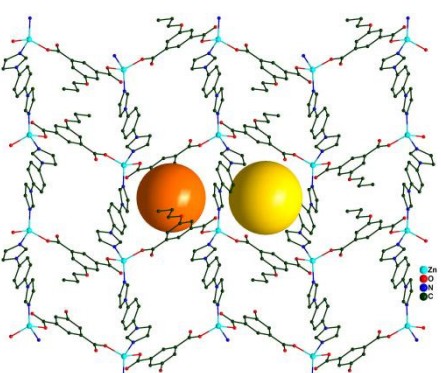

**Figure 2.** View of the 3D framework of complex **1** parallel to the *a*-axis.

　　TGA curves of complex **1**, 5-FU, and 5-FU@**1a** are shown in Figure 4. The relevant experiments for the samples were performed in a N$_2$ atmosphere, where the sample was heated to 800 °C at a rate of 10 °C/min. Complex **1** has thermal stability, as no strictly clean weight loss step occurs below 280 °C. The sharp weight loss above 280 °C corresponds to the decomposition of the framework.

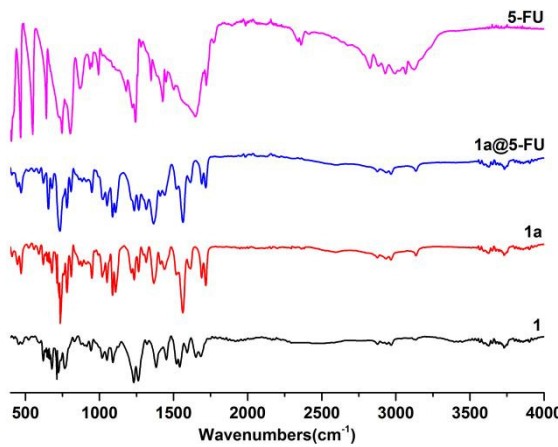

**Figure 3.** IR spectra of complex **1**, complex **1a**, 5-FU@**1a**, and 5-FU.

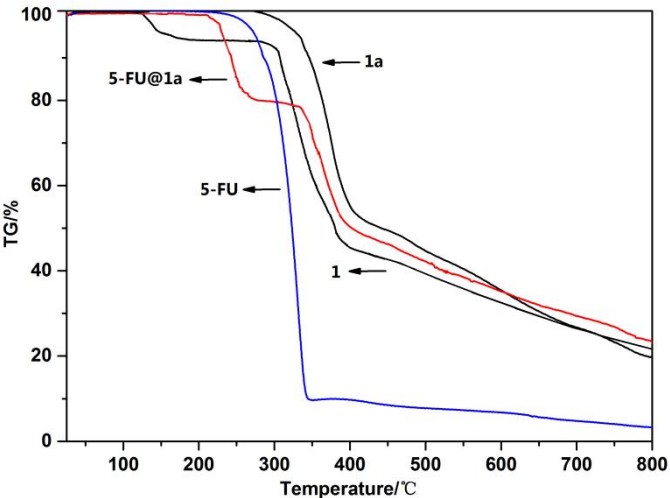

**Figure 4.** TGA curves of complex **1**, complex **1a**, 5-FU, and 5-FU@**1a**.

The water stability of complex **1** was explored through XRPD (Figure 5), and the results show that complex **1** is stable in water and PBS solutions. Furthermore, the bulk samples of **1** and **1a** samples have been characterized by PXRD, where the experimental pattern matches well with the simulated one, showing good purity of the as-synthesized and activated materials.

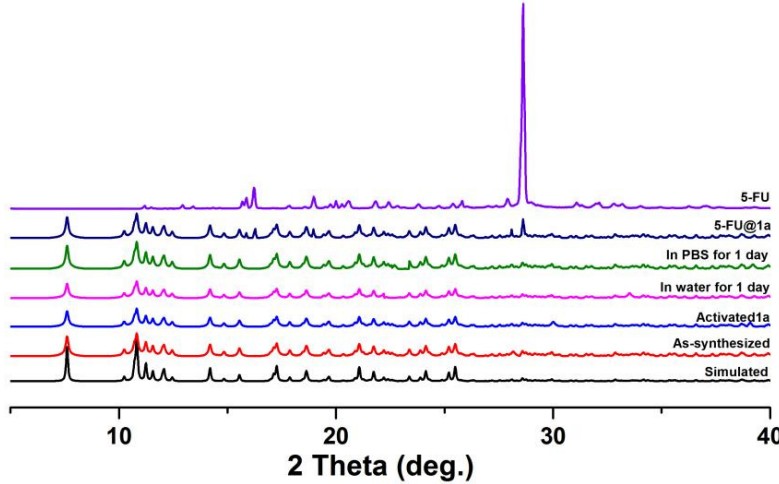

**Figure 5.** PXRD patterns for complex **1**.

In addition, $N_2$ adsorption measurements were used to examine the porosity of the MOF samples after soaking in water for one day. Complex **1** was activated (**1a**) at 120 °C for 8 h under vacuum. The $N_2$ absorption isotherm measurement was performed at 77 K and shows a type I isotherm characteristic of a microporous material (Figure 6a) with a BET surface area of 175 $m^2$ $g^{-1}$. No significant loss in the BET surface area was observed (from 175 to 158 $m^2$ $g^{-1}$) for complex **1** after soaking in water for one day, indicating the maintenance of the porous structures of complex **1** in water. BET-based specific surface area and total pore volume (at $p/p_0 = 0.99$) of complex **1a** were 175 $m^2$/g and 0.201 $cm^3$/g, respectively. A free volume of 19.4% was in good agreement with the theoretically calculated result of 20.1%. The pore size distribution curve had two sharp peaks, indicating complex **1a** has bimodal pore size distributions with mean sizes of 5.0 and 7.9 Å, respectively (Figure 6b).

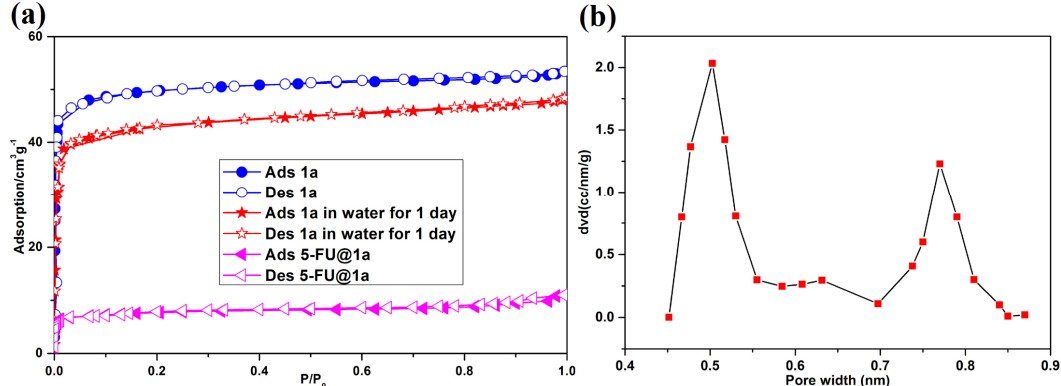

**Figure 6.** (**a**) $N_2$ adsorption-desorption isotherms of activated **1a**, **1a** after in water for one day, and 5-FU-@**1a** at 77 K. (**b**) Pore size distribution of complex **1a** based on the NLDFT model.

## 2.3. Drug Delivery of **1a** for 5-FU

Complex **1** shows a rhombus-like channel ($4.8 \times 13.8$ $Å^2$ and $10.0 \times 8.3$ $Å^2$) is suitable for the loading of 5-FU ($5.3 \times 5.0$ $Å^2$) [28]. After the adsorption of 5-FU, the results obtained were verified using IR and TGA analysis (Figures 3 and 4). The features at 1684 and 1383 $cm^{-1}$ are associated with the -O-C-O- groups. The strong band at 1260 $cm^{-1}$ is attributed to the fluorine atom of 5-FU [29]. The bands in the range from 800–550 $cm^{-1}$ may be due to the C-F bond [28]. The bands at about 3626 $cm^{-1}$ and 3576 $cm^{-1}$ may be attributed to the N-H stretching of 5-FU. The adsorptive capacity of complex **1a** for 5-FU was tested using UV-vis. A maximum 5-FU loading was achieved by evaluating different 5-FU to porous solid ratios and contact times (Table 1). The loading of 5-Fu onto complex **1a** at mass ratios of 1:1, 1:3, 1:5, 3:1, 5:1, 7:1, 10:1, and 15:1 for 3 days is shown in Table 1. The 5-FU loading onto complex **1a** at a mass ratio of 10:1 was best. The effect of loading time on the loading rate of 5-FU by complex **1a** was also investigated under the condition of a 10:1 mass ratio, and three sets of parallel experiments were conducted in A, B, and C. As shown in Figure 7, the loading rate reached its maximum value at 24 h, decreased after 36 h, and almost stabilized after 48 h. The decrease of the loading rate at 36 h was probably due to the adsorption of 5-FU on the surface of complex **1a**, and the loading rate decreased with the increase in stirring time. The decrease in the loading rate at 36 h may be due to the adsorption of 5-FU onto the surface of complex **1a** and the decrease in the loading rate may be due to the desorption of 5-FU as the stirring time increased. In conclusion, using a weight ratio of 10:1 (5-FU:complex **1a**) and soaking complex **1a** for 24 h in 50 mL of methanol produced the best results. Furthermore, the BET surface area of the 5-FU loaded MOF is 23 $m^2$ $g^{-1}$, which shows an 86.8% reduction as compared with the activated complex **1a** samples (175 $m^2$ $g^{-1}$).

**Table 1.** The encapsulation efficiency of 5-FU affected by the 5-FU/material weight ratio.

| 5-FU:1a | 1:1 | 1:3 | 1:5 | 3:1 | 5:1 | 7:1 | 10:1 | 15:1 |
|---|---|---|---|---|---|---|---|---|
| Contact times (days) | 1/2/3/4 | 1/2/3/4 | 1/2/3/4 | 1/2/3/4 | 1/2/3/4 | 1/2/3/4 | 1/2/3/4 | 1/2/3/4 |
| Encapsulation efficiency (wt%) | 6.8/7.6/ 8.5/8.0 | 7.9/7.3/ 7.8/6.2 | 9.4/9.9/ 11.0/8.7 | 11.4/11.9/ 12.5/10.0 | 10.1/11.3/ 12.1/9.4 | 13.4/14.2 /15.4/11.2 | 15.2/16.8 /19.3/12.3 | 11.2/12.5/ 12.8/10.1 |

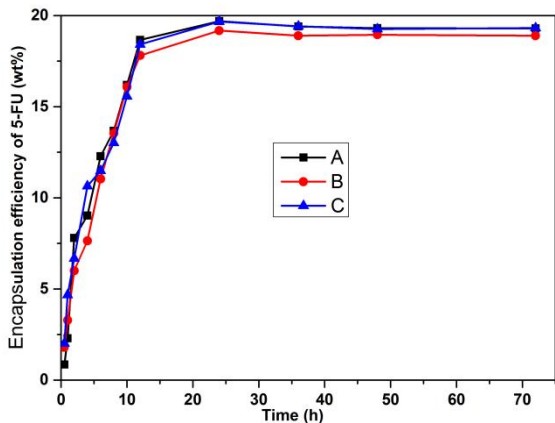

**Figure 7.** The encapsulation efficiency of 5-FU for different times under the 10:1 condition.

The 5-FU@**1a** obtained by loading for 24 h at a mass ratio of 10:1 was added to a conical flask containing 50 mL PBS, allowed to release for 120 h, and 0.1 mL of the solution was pipetted with a pipette at 0, 1, 2, 4, 6, 8, 10, 12, 24, 36, 48, 72, 96, and 120 h in a 10 mL volumetric flask and dispensed to the scale line and left for 5 min before the absorbance was measured by UV spectrophotometer. At the same time, an equal amount of PBS was added to the conical flask, and the concentration of 5-FU after release was calculated according to the linear regression equation with PBS as the solvent, and the release rate of 5-FU was further calculated. As shown in Figure 8, the cumulative release rate was 40.26% at 12 h and 70.05% at 120 h, indicating that 5-FU@**1a** has a slow release performance. Within 20 h, no "burst effect" was observed, while 70% of the 5-FU drug molecules were released after 120 h. There are two different stages of 5-FU release: 40.26% (0–14 h) for the first stage and 29.79% (14–120 h) for the last stage. Drug molecules may be released due to the concentration differences between the dialysis bag and the PBS solution. Compared to these reported MOFs with similar pore sizes, 5-FU within the framework of complex **1a** exhibits a longer release time [21,22,30,31]. The slow release may be due to the strong hydrogen bonds (N-H···O, C-H···O, C-H···F) and π···π stacking interactions between the 5-FU and the channels of complex 1a, in which the free carboxylate oxygen (O2, O5), the coordinated carboxylate oxygen atoms (O1, O4), the hydrogen atoms of imbm ligands, and the five/six element rings from 5-PIA and imbm ligands are pointed to the channels [17].

As a result, the loading efficiency of complex **1a** reached 19.3 wt%, which is close to the value reported in recently published papers [21–23,30,31], but lower than that obtained in other studies [32–34]. TGA and elemental analysis were also used to determine the adsorption amount of 5-FU in complex **1a**. Weight loss of about 2.0% from 120 to 220 may be attributed to the release of water molecules. Then a sharp weight loss above 220 °C may be due to the release of 5-FU. When the temperature is raised to 350 °C, complex **1** starts to decompose. The loading efficiency of complex **1a** was also investigated through elemental analysis, with a value of 20.2 wt%. UV-vis adsorption spectroscopy was used to measure the release of 5-FU drug molecules from 5-FU@**1a** powder soaked in PBS buffer solution.

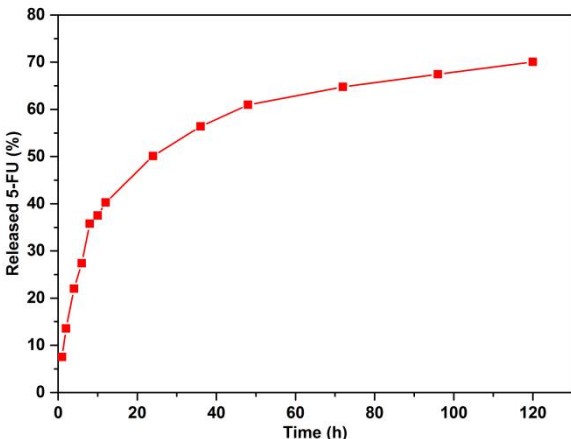

**Figure 8.** Release of 5-FU from the drug-loaded complex **1a** (% 5-FU vs. time).

### 2.4. DFT Simulations

Figure S3 shows the DFT optimized geometry gained from one 5-FU molecule per pore of complex **1a**, which roughly corresponds to the experimental uptake.

### 2.5. MTT and Zebrafish Results

MTT assays were used to determine the viability of NIH-3T3 and HEK-293 cells incubated with complex **1** at different concentrations. When the concentration was below 500 µg/mL for 24 h and 48 h, the survival rate of NIH-3T3 cells was >90%, suggesting that complex **1** had no significant inhibitory effect on the proliferation of NIH-3T3 cells; when the concentrations were less than 5 µg/mL (24 h and 48 h), the survival rate of HEK-293 cells was >90%, and there was no significant inhibition of HEK-293 cell proliferation, but it showed some toxicity between 5–500 µg/mL (Figure 9).

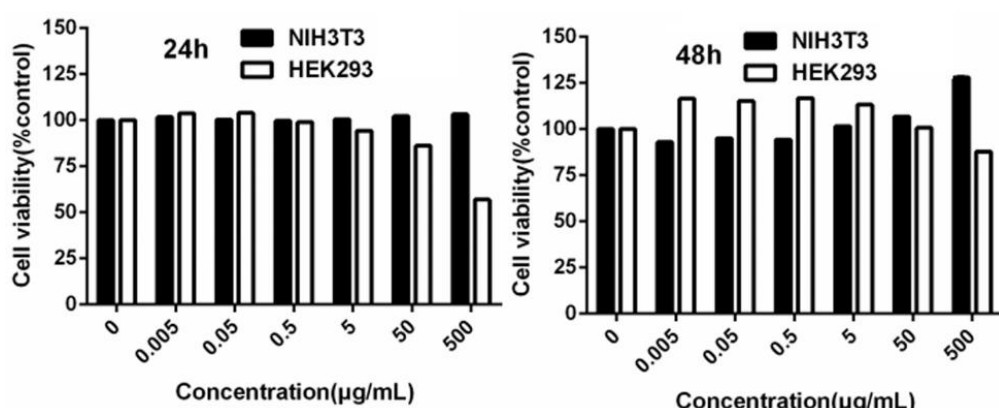

**Figure 9.** MTT toxicity assays of complex **1** on HEK-293 and NIH-3T3 cells at 24 h and 48 h, respectively.

Figure 10 shows that the zebrafish behaved normally overall after 24 h of administration. After 48 h of administration, the vessels, which were fluorescently labeled at the same dose, did not show very significant changes, indicating that the inhibition of vessels by the vector was not significant.

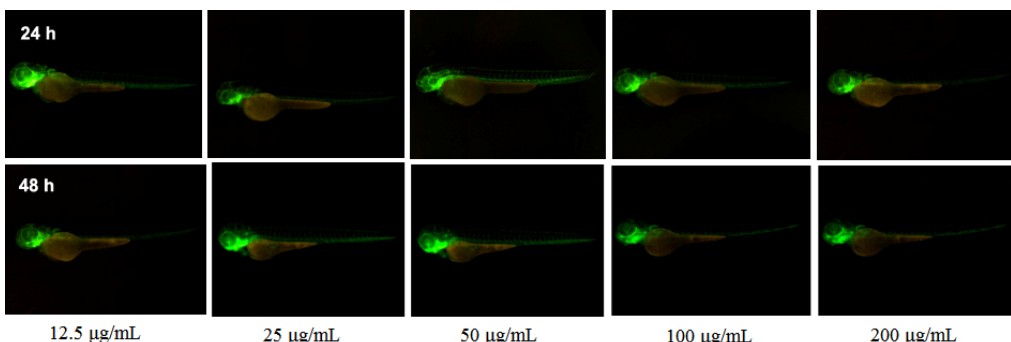

**Figure 10.** The zebrafish toxicity tests of complex **1** under different concentrations.

### 3. Materials and Methods

#### 3.1. Materials and Physical Measurements

Reagent grade 5-propoxy-isophthalic acid (99%), 1,4-di(1H-imidazol-1-yl)benzene (99%), and $Zn(NO_3)_2 \cdot 6H_2O$ metal salt (99.99%) were obtained from Aladdin and used as received. Elemental analyses for C, H, and N were carried out on a Vario ELIII elemental analyzed. IR spectra (KBr) were measured on a Shimadzu 440 spectrometer. Thermal analysis Thermogravimetric (TGA) was carried out in a simultaneous TGA-DTA analyzer DTG-60 Shimadzu instrument. Absorption spectra were determined using the PGEN-ERAL TU-1901 UV-vis Spectrophotometer. Powder X-ray diffraction patterns (PXRD) were performed using a Bruker AXS $D_8$ advance diffractometer with Cu-K$\alpha$ radiation ($\lambda = 1.54186$ Å). Isotherms of $N_2$ in complex **1a** were conducted at 77 K using an automatic volmetric adsorption apparatus (Autosorb-1 manufactured by Quantachrome instruments, Boynton, FL, USA).

#### 3.2. Preparations of Complex **1** and Complex **1a**

5-propoxy-isophthalic (0.067 g, 0.3 mmlo), 1,4-di(1H-imidazol-1-yl)benzene (0.063 g, 0.3 mmol), $Zn(NO_3)_2 \cdot 6H_2O$ (0.089 g, 0.3 mmol), $H_2O$ (6 mL), and ethanol (6 mL) were added to a 23 mL Teflon reactor, and the resulting solution was heated (130 °C, 72 h). The mixture was cooled to room temperature at a rate of 10 °C/h. Colorless crystals were collected with a yield of 51%. Anal. Calcd for $C_{23}H_{24}N_4O_7Zn$ (**1**): C, 51.7; H, 4.5; N, 10.5. Found: C, 52.3; H, 4.3; N, 11.2. IR (KBr pellet) (cm$^{-1}$): 3625 (m), 2964 (w), 2873 (w), 1684 (m), 1653 (m), 1589 (w), 1546 (s), 1514 (s), 1451 (m), 1383 (s), 1260 (vs), 1229 (vs), 1090 (m), 1051 (w), 1015 (w), 943 (w), 765 (w), 710 (s), 622 (s), 519 (w), and 448 (w).

The activated sample of complex **1** (**1a**) was obtained by treating complex **1** under vacuum at 120 °C for 8 h.

The complex **1a** samples, after soaking in water for one day, were treated by drying at 80 °C for 2 h, then further activated at 120 °C for 8 h under vacuum conditions to obtain complex **1a**, which was used for the $N_2$ adsorption isotherm experiment.

#### 3.3. X-ray Crystal Structural Determination

Crystal data for complex **1** were obtained using a Bruker APEX II CCD area-detector diffractometer equipped at 50 kV and 30 mA with MoK$\alpha$ radiation ($\lambda = 0.7107$ Å). The Bruker AXS APEX II software was used for data collection and reduction [35]. The crystal structure of complex **1** was solved through direct methods and refined based on $F^2$ by the full matrix least-squares methods using SHELXTL [36]. All the non-hydrogen atoms were located using the difference Fourier map and refined anisotropically. Hydrogen atoms attached to carbon and oxygen were placed in geometrically idealized positions and refined using the riding model. Guest water molecules with badly disordered electrons were removed by SQUEEZE (PLATON) (A Multipurpose Crystallographic Tool; Utrecht University: Utrecht, The Netherlands, 2005) [37]. By combining elemental analyses with infrared and thermogravimetric characterizations, we arrived at formula unit complex **1**.

Crystal data, as well as details of data collection and refinement, are shown in Table 2. Selected bond lengths and bond angles of complex **1** are shown in Table 3.

**Table 2.** Crystallographic data of complex **1**.

| Formula | $C_{23}H_{20}N_4O_5Zn$ | Z | 4 |
|---|---|---|---|
| Formula weight | 497.80 | Density (g cm$^{-3}$) | 1.281 |
| Cystal system | monoclinic | Limiting indices | $-12 < h < 12$, $-21 < k < 15$, $-21 < l < 19$ |
| Space group | P2$_1$/n | Reflections collected/unique | 5828/3917 |
| Temperature (K) | 296.15 | R$_{int}$ | 0.0421 |
| Size (mm) | 0.21 × 0.17 × 0.15 | F(000) | 1024 |
| a (Å) | 9.5699(9) | θ (°) | 2.468–27.587 |
| b (Å) | 16.5045(16) | Goodness-of-fit on $F^2$ | 0.988 |
| c (Å) | 16.4350(16) | $R$ ($I > 2\sigma$) | $R_1 = 0.0431$ w$R_2$ =0.0951 |
| β (°) | 96.097(2) | $R$ (all data) | $R_1 = 0.0768$ w$R_2$ =0.1074 |
| V (Å$^3$) | 2581.2(4) | Largest diff. Peak and hole (Å$^{-3}$) | 0.33, $-0.37$ |

**Table 3.** Selected bond length and angles of complex **1** (Å, °).

| Bond Lengths | Value (Å) | Bond Angles | Value (°) |
|---|---|---|---|
| Zn1-O1 | 1.9359(17) | O1-Zn1-O4 [i] | 109.40(8) |
| Zn1-O4 [i] | 1.9534(17) | O1-Zn1-N1 | 111.08(8) |
| Zn1-N1 | 1.989(2) | O1-Zn1-N4 [ii] | 98.37(8) |
| Zn1-N4 [ii] | 2.021(2) | O4 [i]-Zn1-N1 | 121.65(8) |
| | | O4 [i]-Zn1-N4 [ii] | 105.97(9) |
| | | N1-Zn1-N4 [ii] | 107.67(9) |

Symmetry codes: i = x + 1/2,−y + 1/2,z + 1/2; ii = −x + 5/2,y − 1/2,−z + 3/2.

### 3.4. Loading of 5-FU

The 5-FU solutions were prepared with methanol and PBS (pH = 7.4) at a concentration of 8 μg/mL$^{-1}$, respectively, and the methanol/PBS mized solution (pH = 7.4) was used as the reference solution as a blank control. The results were scanned in the wavelength range of 190–400 nm, as shown in Figure S1. It was shown that the maximum absorption wavelength was 265 nm for both methanol and PBS (pH = 7.4).

Five gradient standard solutions of 5-FU (10 mg) with different concentrations (10 μg/mL, 20 μg/mL, 30 μg/mL, 40 μg/mL, and 50 μg/mL) were prepared using methanol and PBS (pH = 7.4) as solvents, respectively, and the methanol/PBS solution was used as blank control to determine the absorbance at the absorption wavelength of 265 nm. The linear relationship between absorbance (A) and concentration (c) at this wavelength was established. The linear relationship between absorbance (A) and concentration (c) at this wavelength was established as shown in Figure S2. The linear regression equations for different solvents are shown in Table 3. The results showed that the linear relationship was good for the mass concentration of 5-FU in the range of 10-50 μg/mL$^{-1}$, and the linear regression equation was used to calculate the concentrations of 5-FU after loading and release.

5-Fluorouracil (5-FU) (99%, Aladdin) loading was carried out using the methanol solution. 5-FU loading as follows: complex **1a** (about 50 mg) was dispersed in a 50 mL methanol solution in a volumetric flask; add methanol to the scale; shake well; and let stand for 10 min. The solution was placed in a 100 mL conical flask, and the mass ratios of 5-FU:1a were set at 1:1, 1:2, 1:3, 1:5, 3:1, 5:1, 7:1, 10:1, and 15:1. The corresponding amount of complex **1a** was weighted and added to the 100 mL conical flask. The conical flask was stirred at 350 r min$^{-1}$ for 72 h in a magnetic stirrer, and 0.1 mL of the solution was pipetted with a pipette at 0, 0.5, 1, 2, 4, 6, 8, 10, 12, 24, 36, 48, 60, and 72 h in a 10 mL volumetric flask

and dispensed to the scale line. The absorbance was measured by UV-Vis after standing for 5 min. The maximum 5-FU@**1a** was obtained after the solution was centrifuged, washed with methanol, and dried at 70 °C for 4 h, and the result was obtained based on Equation 1. Each test was repeated four times in parallel. The amount of 5-FU incorporated in the complex **1a** powder was investigated by UV-Vis absorption spectroscopy based on the 5-FU unique absorbance peak in the UV-Vis region at 265 nm.

Complex **1a** before and after the 5-FU entrapment was characterized by IR, PXRD, and TGA.

$$5\text{-}FU \; Encapsulation \; Efficiency \; (\text{wt\%}) = \frac{5\text{-}FU \, loaded \, 1a \, (\text{mg})}{Total \; amount \, (\text{mg})} \times 100\% \qquad (1)$$

### 3.5. Release of 5-FU

5-FU@**1a** (50 mg) was dispersed in a phosphate buffer solution (PBS) (2 mL, pH 7.4, 37 °C) and then poured into a dialysis bag. It was then incubated in PBS (60 mL, pH 7.4, 37 °C) and shaken at 60 rpm for at least 4 h. A 2 mL aliquot of the solution was collected at different time intervals (0, 1, 2, 4, 6, 8, 10, 12, 24, 36, 48, 72, 96, and 120 h), and the concentration of 5-FU was tested by UV-Vis adsorption spectroscopy at an excitation wavelength of 265 nm.

### 3.6. Simulation

The Dmol$^3$ code of the Material Studio software was used to calculate the geometry optimization of the 5-FU molecule under vacuum conditions [38]. During the simulated annealing, the 5-FU molecule is able to explore the lowest-energy conformation with a continuously decreasing temperature [39]. Then a DFT geometry optimization procedure was used to achieve the performance of 5-FU within the **1a**. During the simulation, **1** is fixed, and the 5-FU molecule is flexible and can move around. The PBE GGA density function was used in the above calculations [40,41].

### 3.7. Cytotoxicity Assay

In vitro cytotoxicity MTT assays of complex 1 were performed using the NIH-3T3 and HEK-293 cell lines. Firstly, cells were cultured for 24 h in the 96-well plate with a cell density of 5000 cells per well at 37 °C with 5% $CO_2$. Then, the culture medium was replaced by fresh DMEM containing complex **1** at various concentrations. After incubation for 48 h, MTT solution was added to each well to a final concentration of 0.5 mg/mL per well, and the cells were incubated further for 4 h at 37 °C. After the media was removed, the cells were then dissolved in DMSO solution and quantified through absorption spectrophotometry at 490 nm with a Bio-Rad 3500 microplate reader (Bio-Rad, Hercules, CA, USA). Each experiment was operated three times, and the values were adopted. All the statistical analyses were calculated by SPSS 11.0 with the data of mean ± SD. The significant difference between the experimental and control groups was carried out by the T-test method and identified by a $p < 0.05$. The cell viability was obtained as follows:

$$\text{Cell viability (\%)} = [(OD_1 - OD_3)/(OD_2 - OD_3)] \times 100$$

wherein $OD_1$, $OD_2$, and $OD_3$ are the optical densities of the cell culture with sample, without sample, and of the medium, respectively.

### 3.8. Tests of Zebrafish

The selection of Zebrafish culture and embryos were gained from JiNan University (Guangzhou, China) [42]. The preparation suspension of complex **1** and the exposure experiment by zebrafish of complex **1** followed the reported references [21,43]. Six hpf zebrafish embryos with normal development were picked under a stereomicroscope and given 20.0 mg/L, 10.0 mg/L, 5.0 mg/L, 2.5 mg/L, and 1.25 mg/L of complex **1a** solution according to each group of 25 embryos, with egg water solution as a negative control

group. After administration, zebrafish embryos were placed in an incubator at 28.5 °C. Observations were made daily and regularly. One drop of 3% methylcellulose was placed on a fluted slide. After the zebrafish embryos were de-membranated or stripped, the zebrafish were anesthetized by dropping 1 × (0.16 mg/mL) MS-222. Aspirate the zebrafish with a pipette, place it vertically until the zebrafish sinks to the mouth of the tube, and gently squeeze it slightly to drop it onto the surface of methylcellulose. Pluck the zebrafish with a hair band and adjust its posture to make it suitable for observation.

If taking pictures sideways, try to make the eyes and body segments on both sides overlap, and ensure the tail and body are at the same level. After the zebrafish is positioned, observe and take pictures under a body vision microscope.

### 3.9. Live Subject Statement

This experimental protocol involved the butchering of zebrafish embryos, an action that was in accordance with the National Institute of Health's Guide for the Care and Use of Laboratory Animals, China.

### 4. Conclusions

In brief, we prepared a new 2D zinc-based MOF involving 5-propoxy-isophthalate and 1,4-di(1H-imidazol-1-yl)benzene mixed ligands with 4.8 × 13.8 and 10.0 × 8.3 Å$^2$ sized channels along the *a* axis, respectively. The N$_2$ adsorption isotherm of complex **1a** shows a type I isotherm characteristic of a microporous material with a BET surface area of 175 m$^2$ g$^{-1}$. The pore size distribution curve of complex **1a** shows two sharp peaks, meaning complex **1a** has two pore size distributions of sizes 5.0 and 7.9 Å, respectively. MTT assays and zebrafish experiments demonstrated that title compounds are nontoxic for NIH-3T3 and HEK-293 cell lines at certain concentrations and residence times. In order to explore the encapsulation of complex **1a**, the anticancer drug molecule (5-fluorouracil) was loaded into complex **1a** with a loading amount of ~19.3 wt%. The drug delivery of 5-FU@**1a** was investigated in a PBS solution, and the results showed that 5-FU could be released from complex **1a** in PBS without any "burst effect". About 70% of 5-FU was released from the drug delivery system within 120 h. The results of this study can enrich the study of drug delivery systems based on MOF materials.

**Supplementary Materials:** The following supporting information can be downloaded at: https://www.mdpi.com/article/10.3390/inorganics10110202/s1, Figure S1: UV absorption curves of 5-FU in methanol and PBS, respectively; Figure S2: Linear regression curves of concentration versus absorbance of 5-FU measured at 265 nm. Figure S3: Optimized geometries of 5-FU within the pore of complex **1a**.

**Author Contributions:** Conceptualization, F.L. and L.Q.; methodology, L.Q.; software, Y.L.; validation, J.W., S.G. and M.W.; formal analysis, S.X.; investigation, M.L.; resources, D.M.; data curation, F.L.; writing—original draft preparation, F.L. and L.Q.; writing—review and editing, D.M.; visualization, Y.L.; supervision, F.L.; project administration, D.M.; funding acquisition, D.M. All authors have read and agreed to the published version of the manuscript.

**Funding:** This research was supported by the characteristics of innovative projects in Colleges and Universities in Guangdong Province (No. 2020KTSCX157), the Special Projects in Key Fields of Colleges and Universities in Guangdong Province (No. 2020ZDZX047), the National Undergraduate Innovation and Entrepreneurship Training Project (No. 202110580017, 202110580018), the Guangdong Provincial Key Laboratory of Environmental Health and Land Resource (No. 2020B121201014), and the Guangdong Technology and Equipment Research Center for Soil and Water Pollution Control.

**Data Availability Statement:** The detail crystallographic data of complex **1** are provided in CIF files. CCDC: 2215223. Copies of this information may be obtained free of charge from the Director, CCDC, 12 Union Road, Cambridge, CB21EZ, UK (fax: +44-1223-336-033; e-mail: deposit@ccdc.cam.ac.uk or (http://www.ccdc.cam.ac.uk, accessed on 25 October 2022) or also available from the author upon request.

**Acknowledgments:** It is a pleasure to thank Peng Yan for the characterization of the PXRD results.

**Conflicts of Interest:** The authors declare no conflict of interest.

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
