# Peer review of "A 2D Porous Zinc-Organic Framework Platform for Loading of 5-Fluorouracil"

_inorganics, doi:10.3390/inorganics10110202_

Round 1

Reviewer 1 Report

Dear Editor in this work a Zn based MOF was synthesized and used for encapsulation of an anticancer drug. The paper is not well written and needs substantial revision. Findings are also not well presented.

Introduction is no informatic concerning what authors have done and maybe the important findings and should be rewritten.

Introduction needs also to be rewritten since only few informations about the use of MOFs as anticancer carriers, their toxicity, their different structures, etc., was added.

In introduction was reported that MOFs are promising drug de
livery carriers for anti-cancer delivery or generally biomedical applications but the literature to support this is limited. I propose to add some references there (5-6 added papers) to strength this (please see a paper Int. J. Pharm. 509; 208–218: 2016).

There is no information about the novelty of the synthesized MOF. Was it for first time synthesized? Why was this drug chosen? What problems authors try to solve? All these should be clarified in introduction or in results and discussion.

What is the porosity of the synthesized MOF? This is very important for drug absorption.

It is not clear if the drug was absorbed in MOF’s pores or in the surface.

What is the physical state of the drug in the MOF carrier? Was encapsulated only one amount (wt%) or authors tried also additional amounts?

Author Response

Dear Editor in this work a Zn based MOF was synthesized and used for encapsulation of an anticancer drug. The paper is not well written and needs substantial revision. Findings are also not well presented.

Introduction is no informatic concerning what authors have done and maybe the important findings and should be rewritten.

Respond: The introduction has been rewritten following the suggestions.

Introduction needs also to be rewritten since only few informations about the use of MOFs as anticancer carriers, their toxicity, their different structures, etc., was added.

Respond: The introduction has been rewritten following the suggestions.

In introduction was reported that MOFs are promising drug de

livery carriers for anti-cancer delivery or generally biomedical applications but the literature to support this is limited. I propose to add some references there (5-6 added papers) to strength this (please see a paper Int. J. Pharm. 509; 208–218: 2016).

Respond: The introduction has been rewritten following the suggestions. Furthermore, several related references have been cited.

There is no information about the novelty of the synthesized MOF. Was it for first time synthesized? Why was this drug chosen? What problems authors try to solve? All these should be clarified in introduction or in results and discussion.

Respond: Good questions. Title compound was first synthesized and used for drug delivery. The 5-FU was chose for drug delivery due the molecule size of 5-FU is suitable for loading into the channel of 1a.In fact, we have also tried other drug molecules (for example, Busulfan, Cisplatin), but the results is not satisfactory.

What is the porosity of the synthesized MOF? This is very important for drug absorption.

Respond: Good questions. The N2 sorption isotherm of 1a has been carried out and the results shown that the BET of 1a is 175 m2/g with two pore sizes (5.0 and 7.9 Å), as shown in Figure 6.

It is not clear if the drug was absorbed in MOF’s pores or in the surface.

Respond: Good question. In fact, operations such as centrifugation and rinsing are intervened to ensure that drug molecules are not loaded on the surface. Furthermore, the N2 sorption isotherm of 5-FU@1a compounds was also investigated to ensure that the 5-FU was adsorbed into the framework pores (Figure 6).

What is the physical state of the drug in the MOF carrier? Was encapsulated only one amount (wt%) or authors tried also additional amounts?

Respond: Good questions. We have added the DFT simulation of 5-FU (Figure S3). We have tired different amount (wt%), which as shown in Table 3.

Reviewer 2 Report

        This work synthesized a new 2D zinc-based MOF with an exceptional chemical stability in water and PBS, and utilized the MOF for drug uptake and release. The work is well organized, and the novelty of this work is high. Given the new MOF structure and the attractive properties, this work should be of interest to the readership of the inorganic chemistry community. Thus I highly recommend the publication of this work in Inorganics. However, some issues are there and should be addressed before publication.

1. Purity of each chemical used here is missing in section 4.1.

2. The Supplementary Material file is not there.

3. I assumed that the material “1a” indicates activated powder of “1”, right? Then the authors should provide the detailed experimental procedure regarding how to activate complex 1 to obtain 1a. The current version just stopped at the cooling step after crystal growth.

4. The porosity of 1a and 5-FU@1a should be investigated by nitrogen adsorption-desorption experiments, and the porosity should be reported.

5. What is the possible interaction between 5-FU and complex 1? I suggest the authors to add some comments and discussion on this point.

6. Since the MOF here is highly stable in PBS, how did 5-FU get released when PBS is present? I suggest the authors to add some discussion on the possible reasons for the release of 5-FU from the MOF in PBS. How about just using pure water for this release experiment?

Author Response

This work synthesized a new 2D zinc-based MOF with an exceptional chemical stability in water and PBS, and utilized the MOF for drug uptake and release. The work is well organized, and the novelty of this work is high. Given the new MOF structure and the attractive properties, this work should be of interest to the readership of the inorganic chemistry community. Thus I highly recommend the publication of this work in Inorganics. However, some issues are there and should be addressed before publication.

  1. Purity of each chemical used here is missing in section 4.1.

Respond: We have added the purity of each chemical.

  1. The Supplementary Material file is not there.

Respond: We have uploaded the supplementary materials.

  1. I assumed that the material “1a” indicates activated powder of “1”, right? Then the authors should provide the detailed experimental procedure regarding how to activate complex 1 to obtain 1a. The current version just stopped at the cooling step after crystal growth.

Respond: We have the detail experimental procedure how to activate 1 to obtain 1a in the last of introduction section.

  1. The porosity of 1a and 5-FU@1a should be investigated by nitrogen adsorption-desorption experiments, and the porosity should be reported.

Respond: Good questions. The N2 sorption isotherm of 1a has been carried out and the results shown that the BET of 1a is 175 m2/g with two pore sizes (5.0 and 7.9 Å), as shown in Figure 6.

  1. What is the possible interaction between 5-FU and complex 1? I suggest the authors to add some comments and discussion on this point.

Respond: We have added the comments and discussion in the paper (2.3 section)

  1. Since the MOF here is highly stable in PBS, how did 5-FU get released when PBS is present? I suggest the authors to add some discussion on the possible reasons for the release of 5-FU from the MOF in PBS. How about just using pure water for this release experiment?

Respond: Very good questions. The detail drug release experiment was described in ‘4.5. Release of 5-FU’. The 5-FU drug molecules are dissociated due to the concentration differences and shaking at 60rpm. We have added the discussion in the paper (2.3 section). The choice of studying the release behavior in PBS solution is to simulate the human blood environment in order to provide experimental data for subsequent further application studies.

Reviewer 3 Report

The manuscript by Liang Qin et al entitled "A 2D porous zinc-organic framework platform for loading of 5- 2 Fluorouracil" report the synthesis and characterization of 2-D MOF with the loading of 5- 2 Fluorouracil was also performed. In my opinion manuscript can be accepted after major revision.

1. Please rewrite the abstract according to your results.

The introduction parts lacks sound literature on the MOF materials. Please see the following articles Coordination Chemistry Reviews 474 (2023): 214859, Journal of Solid State Chemistry 315 (2022): 123482, International Journal of Pharmaceutics (2022): 122228, Journal of Solid State Chemistry (2022): 123602, Electrochemical Applications of Metal-Organic Frameworks, pp. 17-35. Elsevier, 2022, Dalton Trans., 2022,51, 14817-14832

I do not agree with the author claims that the MOF is porous in nature. Please perform BET for the same.

I am also curious about the high stability of MOF. Can author perform some additional experiments to established the high stability of MOF.

The experimental results can be compare with some recent literature.

The conclusion is very poorly written. The author needs to improve the english of the manuscript.

Author Response

The manuscript by Liang Qin et al entitled "A 2D porous zinc-organic framework platform for loading of 5- 2 Fluorouracil" report the synthesis and characterization of 2-D MOF with the loading of 5- 2 Fluorouracil was also performed. In my opinion manuscript can be accepted after major revision.

  1. Please rewrite the abstract according to your results.

The introduction parts lacks sound literature on the MOF materials. Please see the following articles Coordination Chemistry Reviews 474 (2023): 214859, Journal of Solid State Chemistry315 (2022): 123482, International Journal of Pharmaceutics (2022): 122228, Journal of Solid State Chemistry (2022): 123602, Electrochemical Applications of Metal-Organic Frameworks, pp. 17-35. Elsevier, 2022, Dalton Trans., 2022,51, 14817-14832

Respond: The introduction section have been written following the suggestions. In addition, above mention references have been cited in the paper.

I do not agree with the author claims that the MOF is porous in nature. Please perform BET for the same.

Respond: Good question. The N2 sorption isotherm of 1a has been carried out and the results shown that the BET of 1a is 175 m2/g with two pore sizes (5.0 and 7.9 Å), as shown in Figure 6.

I am also curious about the high stability of MOF. Can author perform some additional experiments to established the high stability of MOF.

Respond: Good question. The N2 sorption isotherm of 1a after soaking in water for one day has also been investigated to established the high stability of this MOF.

The experimental results can be compare with some recent literature.

Respond: Good question. The experimental results of drug uptake and release have been compared with some recent literature (Ref. 34-36)

The conclusion is very poorly written. The author needs to improve the english of the manuscript.

Respond: We have improved the paper following the suggestions.

Round 2

Reviewer 1 Report

Dear Editor, in the revised manuscript all my comments have been well  addressed . So, I propose to accept  the paper for publication.

Author Response

Thank you for your recognition of our work.

Reviewer 2 Report

Comments were mostly addressed. The paper can be accepted after fixing the following minor points:

1. Regarding how to activate 1 to obtain 1a, this experimental detail should be included in the Experimental section rather than the end of Introduction.

2. Figure 6a: There are many isotherms, but the Figure caption does not reflect the content of this Figure. The authors should correct the Figure caption.

3. Figure 6a: How to re-activate the material after treating with water for 1 day? The authors are suggested to add the experimental details in the experimental section. 

Author Response

1.Regarding how to activate 1 to obtain 1a, this experimental detail should be included in the Experimental section rather than the end of Introduction.

Respond: We have added the detail experimental in ‘4.2 Preparations of complex 1 and 1a’ section.

2.Figure 6a: There are many isotherms, but the Figure caption does not reflect the content of this Figure. The authors should correct the Figure caption.

Respond: We have corrected them.

3.Figure 6a: How to re-activate the material after treating with water for 1 day? The authors are suggested to add the experimental details in the experimental section.

Respond: We have added the detail experimental in ‘4.2 Preparations of complex 1 and 1a’ section.

Reviewer 3 Report

The author has put lot of efforts to revise the manuscript. In my opinion the manuscript can be accepted in its current form.

Author Response

(The authors gave the same response as above.)
